# New Insights into Testosterone Biosynthesis: Novel Observations from HSD17B3 Deficient Mice

**DOI:** 10.3390/ijms232415555

**Published:** 2022-12-08

**Authors:** Ben M. Lawrence, Liza O’Donnell, Lee B. Smith, Diane Rebourcet

**Affiliations:** 1College of Engineering, Science and Environment, The University of Newcastle, Callaghan, NSW 2308, Australia; 2Office for Research, Griffith University, Southport, QLD 4222, Australia; 3MRC Centre for Reproductive Health, The Queen’s Medical Research Institute, University of Edinburgh, Edinburgh EH16 4TJ, UK

**Keywords:** androgens, testosterone, HSD17B3, enzymes, canonical pathway

## Abstract

Androgens such as testosterone and dihydrotestosterone (DHT) are essential for male sexual development, masculinisation, and fertility. Testosterone is produced via the canonical androgen production pathway and is essential for normal masculinisation and testis function. Disruption to androgen production can result in disorders of sexual development (DSD). In the canonical pathway, 17β-hydroxysteroid dehydrogenase type 3 (HSD17B3) is viewed as a critical enzyme in the production of testosterone, performing the final conversion required. HSD17B3 deficiency in humans is associated with DSD due to low testosterone concentration during development. Individuals with *HSD17B3* mutations have poorly masculinised external genitalia that can appear as ambiguous or female, whilst having internal Wolffian structures and testes. Recent studies in mice deficient in HSD17B3 have made the surprising finding that testosterone production is maintained, male mice are masculinised and remain fertile, suggesting differences between mice and human testosterone production exist. We discuss the phenotypic differences observed and the possible other pathways and enzymes that could be contributing to testosterone production and male development. The identification of alternative testosterone synthesising enzymes could inform the development of novel therapies to endogenously regulate testosterone production in individuals with testosterone deficiency.

## 1. Background

Androgens are steroid hormones critical for male sexual development, masculinisation, spermatogenesis, and general lifelong male health [1,2,3]. Disruptions to androgen production at any point during life can result in complications contributing to lower quality of life and premature death [2,3].

Androgen biosynthesis is a complex process that occurs in the highly specialised Leydig cells, in the interstitial space of the testis. Androgen biosynthesis involves a succession of enzymatic reactions converting steroid precursors to the biologically active androgens testosterone and the more potent dihydrotestosterone (DHT). Like all steroid hormones, cholesterol is the initial precursor required to make androgens. Therefore, a constant supply is needed for the Leydig cells. Free cholesterol can be derived from (i) de novo cholesterol synthesis, (ii) from the hydrolysis of stored cholesterol esters in lipids, or (iii) from lipoproteins circulating in the serum. It has been demonstrated in mice and rats that the preferred source of cholesterol for steroidogenesis is de novo synthesis [4,5]. Sourcing free cholesterol allows the Leydig cells to begin the process of synthesising androgens.

The first step in steroidogenesis is the conversion of cholesterol into pregnenolone and this process occurs in the inner membrane of the mitochondria of the Leydig cells [6,7]. The binding of luteinising hormone (LH) to the luteinising hormone/choriogonadotropin receptor (LHCGR) causes an increase in the phosphorylation of the steroidogenic acute regulatory (StAR) protein which regulates the transfer of cholesterol to the inner mitochondrial membrane. Cholesterol side-chain cleavage enzyme, CYP11A1 (P450scc), then converts cholesterol into pregnenolone (Figure 1) [7]. Pregnenolone leaves the mitochondria by passive diffusion and all subsequent androgen biosynthesis steps occur in the smooth endoplasmic reticulum of the cell [8].

Within the canonical pathway of androgen biosynthesis (Figure 1), pregnenolone and progesterone are converted to 17OH-pregnenolone and 17OH-progesterone, respectively, by the 17α-hydroxylase activity of CYP17A1 [9]. The 17,20 lyase activity of CYP17A1 then converts 17OH-pregnenolone and 17OH-progesterone into DHEA and androstenedione, respectively [9,10]. HSD17B3, which is exclusively expressed in the testes, can convert DHEA into androstenediol, or androstenedione into the biologically active androgen testosterone. HSD3B1/2 can convert pregnenolone, 17OH-pregnenolone, DHEA and androstanediol into progesterone, 17OH-progesterone, androstenedione, and testosterone, respectively.

The canonical pathway can be divided into the Δ4 and Δ5 pathways, with primary usage depending on animal species (Figure 1). Humans and primates predominantly use the Δ5 route for the conversions required from pregnenolone through to DHEA, with less contribution to the Δ4 route. This is due to the human 17,20 lyase activity of CYP17A1, which has a significantly higher preference towards 17OH-pregnenolone over 17OH-progesterone [10,11]. In contrast, rodents primarily utilize the Δ4 pathway as their CYP17A1 has a stronger affinity towards 17OH-progesterone [10,11].

Free testosterone diffuses into cells and binds to the androgen receptor (AR) to induce androgen-dependent genomic and non-genomic pathways [12]. It can also be transported in circulation by the carrier proteins, sex hormone binding globulin (SHBG) in humans [13], and androgen binding protein in rodents [14]. These proteins transport testosterone to androgen target organs where it can then be converted by steroid-5alpha-reductase (SRD5A) enzymes to DHT. The enzyme that produces DHT is dependent on the tissue as SRD5A1 is the major isoform detected in non-reproductive tissues, such as the liver and skin, whereas SRD5A2 is predominantly expressed in the reproductive organs [15]. Both SRD5A enzymes catalyse this reaction but differ in their substrate affinity and other biochemical properties [16].

While HSD17B3 is the critical enzyme for testosterone production, studies in mice have shown it is not expressed by fetal Leydig cells, and instead is expressed by fetal Sertoli cells during sexual development [17,18]. Therefore, in the mouse, fetal Leydig cells are responsible for producing the androgen precursor, androstenedione, and the fetal Sertoli cells are responsible for catalysing the conversion to testosterone [18]. During postnatal development however, Sertoli cells lose the expression of HSD17B3 and immature Leydig cells differentiate into adult Leydig cells [18]. Leydig cell differentiation results in the expression of steroidogenic enzymes, including HSD17B3, enabling the adult Leydig cells to become the primary site of testosterone synthesis under gonadotrophin stimulation from puberty and throughout adulthood [19,20].

## 2. Human HSD17B3 Deficiency

HSD17B3 deficiency is the most common disorder of androgen synthesis, with 70 reported mutations in humans that result in a disorder of sexual development (DSD) phenotype [21,22]. This disorder is caused by a genetic mutation in the *HSD17B3* gene and results in perturbed sexual differentiation in males as it prevents the sufficient reduction of androstenedione into testosterone. Whilst genetic testing can confirm *HSD17B3* loss-of-function mutations, hormone profiling is used as a diagnostic hallmark, where abnormally high androstenedione to testosterone ratios is seen, with androstenedione being 10 times higher than normal, indicative of reduced HSD17B3 function [21,23,24,25].

46,XY DSD humans with a loss-of-function mutation in *HSD17B3* present with external genitalia that are under-masculinised, appearing as either ambiguous or as female, with a blind vaginal pouch. As a result of the human body’s default setting to develop female external structures without appropriate androgen action, many individuals with this condition are raised as female and diagnosis is often missed during infancy. Interestingly, internally testes are present, which are commonly smaller, undescended (cryptorchidism) and located in the inguinal or intraabdominal regions [21]. Wolffian structures are also present, including epididymis, vas deferens and seminal vesicles. As the Wolffian structures are androgen-dependent, this indicates that low levels of testosterone continue to be produced [21].

One of the more curious observations seen in individuals with HSD17B3 deficiency is that they, to some extent, undergo late-onset virilisation during puberty and it is at this time when most cases are diagnosed. At time of puberty, 46,XY individuals raised as female may seek medical attention due to amenorrhea or due to virilisation. The extent of virilisation at puberty can vary according to the mutation but may include developing male characteristics such as deepening of the voice, heightened body hair, defined male body structure and clitoromegaly, resembling a micropenis [21,25,26,27]. Due to these individuals having some level of androgen action during puberty and low levels of testosterone present, it is suggested that HSD17B3 is not solely responsible for testosterone production in humans during puberty and there may be other 17β-hydroxysteroid dehydrogenase enzymes involved [21]. HSD17B5, also known as AKR1C3, is expressed in peripheral tissues and can be a source of peripheral conversion in some disease states, including prostate cancer [21,28,29]. Therefore, it has been postulated that the circulating testosterone present may be produced in peripheral tissues, rather than in the testes [21]. Another explanation for the late-onset virilisation is that not all HSD17B3 mutations render HSD17B3 completely inactive, and residual HSD17B3 may function to convert low amounts of androstenedione into testosterone, followed by SRD5A enzymes becoming more highly expressed at puberty where they can synthesise DHT [21,30].

Regardless of the extent of virilisation that occurs, HSD17B3-deficient individuals are infertile [31]. This highlights the importance of the human canonical pathway and the role of human HSD17B3 in male sexual development, testosterone production and testis function.

## 3. Transgenic Mouse Models of HSD17B3 Deficiency

Transgenic mouse models can be utilised to dissect androgen biosynthesis, the roles of steroidogenic enzymes, and provide hypotheses for testing in human disorders. The complexities of the steroidogenic pathways (Figure 1) make it challenging to predict outcomes of in vivo models. Development of in silico modelling using the modified Edinburgh pathway notation (mEPN), and the network editing softwares yED and BioLayout, has been useful in displaying the steroidogenesis pathway and for predicting logical hypotheses [32]. These tools have the potential to be used to model different signalling pathways in a multitude of contexts. Both human and rodent steroidogenic pathways in the testis have been modelled using this in silico approach and have been adapted to portray HSD17B3 deficiency [32,33]. In silico modelling of HSD17B3 deficiency in rodents predicted androstenedione production would increase as Leydig cells mature and no testosterone would be produced [32]. Therefore, a similar phenotype in rodents was expected to that seen in humans with HSD17B3 deficiency.

Since the in silico model establishment of the HSD17B3 deficiency in mice, two independent research groups have now reported the phenotypes of independently derived *Hsd17b3*-deficient mouse lines [17,34]. These models were designed to mimic the human mutation causing HSD17B3 deficiency by blocking the conversion of androstenedione to the active androgen testosterone, subsequently causing disruption to the canonical androgen production pathway.

Matching the diagnostic hallmark of HSD17B3 deficiency in human males, and the in silico model, HSD17B3 knockout (KO) male mice had a significantly higher androstenedione to testosterone ratio, indicating that the canonical pathway of androgen production was disrupted [17,34].

Surprisingly though, unlike in humans, male HSD17B3 KO mice remain virilised at birth and can be identified from female littermates [17,34]. Decreased anogenital distance is observed in HSD17B3 KO male mice compared to wild-type males, however, it is not reduced to that seen in females [17,34]. The smaller anogenital distance indicates that HSD17B3 KO mice have reduced, but not absent, androgen action during a key fetal developmental timepoint known as the masculinisation programming window [35].

Further examination of HSD17B3 KO mice revealed that unlike in human males with HSD17B3 deficiency, mice had no major impact on reproductive development [17,34]. Sipila et al. showed a decrease in seminiferous tubule diameter and in testis weight at ~1 and 3 months of age, however, in contrast Rebourcet et al. did not see any testis weight difference in adulthood [17,34]. As seminiferous tubules make up approximately 90% of the testis, a decrease in testis weight is a key indicator of reduced spermatogenesis. Surprisingly though and in contrast to humans, HSD17B3 KO male mice display normal spermatogenesis and remain fertile, something that has widely been accepted to be fundamentally dependent upon the 17-ketosteroid reductase activity of HSD17B3, and its ability to produce testosterone. The decrease in tubule diameter observed by Sipila and colleagues was more prevalent in mice aged 4 weeks [34], prior to the commencement of spermatogenesis, compared to 3-month-old mice, suggesting that the development of the testis may be delayed, rather than spermatogenesis itself. As testosterone is absolutely required for spermatogenesis [36], this indicates that testosterone is being produced to acceptable levels in the HSD17B3 KO mice.

Hormone profiling of HSD17B3 KO adult male mice revealed a marked increase in circulating and intratesticular androstenedione, likely due to a reduced ability to convert this substrate into testosterone [17,34]. Other androgen precursors are also increased in HSD17B3 KO males, including progesterone and 17OH-progesterone, suggesting a backlog of androgen precursors within the pathway [17,34]. Importantly, intratesticular testosterone levels were unchanged, and circulating testosterone and DHT were significantly increased in HSD17B3 KO adult males [17,34]. Circulating LH was also increased, along with increased transcription of *Lhcgr*, *StAR*, *Cyp11a1* and *Cyp17a1* [17,34]. The increased circulating testosterone, LH, and upregulation of steroidogenic biosynthetic enzymes indicate that the hypothalamic-pituitary-gonadal axis, which functions as a negative feedback loop, is dysregulated in these mice. As the hypothalamic-pituitary-gonadal axis is programmed by actions of testosterone and kisspeptin signalling during fetal development, this negative feedback axis may be impacted due to abnormal androgen action during fetal development, however this proposition has not been confirmed [17,37,38].

As HSD17B3 KO males remain fertile, with quantitatively and qualitatively normal spermatogenesis [34], along with a maintenance of intratesticular testosterone [17,34], this suggests the existence of a compensatory mechanism in mice which maintains testosterone production in the absence of HSD17B3. However, the enzyme(s) responsible remains unknown.

The maintenance of testicular testosterone production in the absence of HSD17B3 in mice [17,34] contradicts the predicted testosterone levels portrayed by the in silico model [32], strongly suggesting the existence of other unrecognised androgen biosynthetic enzymes that contribute to testicular testosterone production. These results highlight the utility of transgenic mouse models for identifying and elucidating the roles of specific steroidogenic enzymes and androgens, and suggest that the in silico models of steroidogenesis need more information to accurately predict steroidogenic output in a given context.

The results of Sipila et al. and Rebourcet et al. demonstrate that HSD17B3 does not encapsulate the entirety of testosterone production in mice [17,34]. Other pathways or enzymes, particularly those within the HSD17B family, may influence or be involved in testosterone production, and are potential candidates that could compensate for the loss of HSD17B3 action.

## 4. Possible Explanations for Continued Testosterone Production in HSD17B3-Deficient Mice

### 4.1. The Role of the Adrenal Gland in Androgen Production

Whilst the majority of androgens are produced in the testis, approximately 5% of human androgens are synthesised in the zona reticularis of the human adrenal cortex. This is an alternative source for androgens, such as DHEA and androstenedione, however, these have a low affinity for AR and are considered to be weak androgens [39]. However, these androgen precursors can be transported to other tissues where they are metabolised via the canonical or alternate androgen pathways to produce more potent androgens, specifically testosterone and DHT (Figure 1). The human adrenal cortex can also synthesise 11-oxygenated 19-carbon steroids such as 11β-hydroxyandrostenedione and 11β-hydroxytestosterone utilising the enzyme CYP11B1 [40], however these are modifications of androstenedione and testosterone.

There are species differences between human and mouse adrenal function. In fact, the mouse adrenal is often overlooked as an androgen-producing organ because the CYP17A1 enzyme, important for 17,20 lyase activity in androgen production, is silenced early on in mouse development [41]. Therefore, while the human adrenal gland can produce some androgens, the lack of CYP17A1 in the mouse adrenal greatly reduces its capacity for androgen production [41]. Further, 11-oxygenated steroids that are prevalent in humans, are at undetectable concentrations in mouse circulation [42]. It is however possible that the mouse adrenal gland could produce androgens from circulating testis-derived androstenedione. The adrenal gland expresses the enzyme HSD17B5, which could be involved in converting some of the high circulating androstenedione to testosterone.

To determine if the mouse adrenal gland can compensate for the lack of HSD17B3 by producing testosterone and DHT, Sipila et al. measured androgen concentrations after an adrenalectomy [34]. Surprisingly, the results showed that circulating testosterone and DHT in adrenalectomized HSD17B3 KO mice further increased, ruling out the adrenal gland as the site of continued androgen production [34]. This finding supports observations in human HSD17B3 deficiency cases, as adrenal steroid biosynthesis remains normal in males with HSD17B3 deficiency [43].

### 4.2. Androgen Biosynthesis in Peripheral Tissues

Local conversion of androgens also occurs in peripheral tissues, making them another potential source of androgen production [44]. SRD5A enzymes are expressed in peripheral tissues and can convert testosterone to the more potent androgen DHT.

11β-hydroxysteroids synthesised in the human adrenal gland can be converted to 11-keto androgens, predominantly occurring in peripheral tissues, and these can have androgen activity by acting on AR [45]. There is emerging evidence that the 11-keto androgens, including 11-ketotestosterone and 11-keto-DHT, play a role in male physiology in both humans and mice, however, whether they have a role in postnatal testis function is not established [40,46,47]. Whilst 11-keto-testosterone has been detected in both humans [45] and mice [48], 11-keto-testosterone is another modification of testosterone and therefore does not explain the testosterone synthesis in the HSD17B3 deficient mice [17,34]. However, the role of 11-keto androgens in mice may be amplified and contribute to some of the phenotypic characteristics observed in these mice.

The increased circulating testosterone with unchanged intratesticular levels suggests testosterone synthesis could be occurring in the peripheral tissue of HSD17B3 KO mice. Enzymes expressed in the periphery, such as HSD17B5, are potential candidates that may be responsible. HSD17B5 can convert androstenedione to testosterone and is the major enzyme that performs this conversion in the prostate [28]. HSD17B5 is overexpressed in cancers, including castrate-resistant prostate cancer, resulting in increased testosterone production [44]. Therefore, it can be postulated that HSD17B5 may become overexpressed in peripheral tissues following HSD17B3 ablation.

To further investigate the source of androgens, Sipila and colleagues also performed intra-tissue steroid analysis on the testis and peripheral tissues including epididymis, prostate, adrenal, liver, kidney, adipose tissue, and spleen [34]. These results demonstrated some, although minimal, testosterone levels in peripheral tissues of HSD17B3 KO mice, and high levels in the testis, indicating that testosterone synthesis in the absence of HSD17B3 is likely to be of testicular origin [34].

### 4.3. The Alternate “Backdoor” Pathway

The functionality of the canonical pathway is essential for testosterone production, however, it does not encapsulate the entirety of androgen biosynthesis. In 2003, Wilson et al. discovered a new androgen biosynthesis pathway in the tammar wallaby, where DHT is produced utilising steroid precursors, bypassing the need for testosterone synthesis (Figure 1) [49,50]. Since this discovery, the alternate pathway, commonly known as the backdoor pathway, has been identified in other species including both mice and humans [51,52,53].

Androgen precursors produced in the canonical pathway are converted into the alternate pathway by SRD5A1 and SRD5A2 enzymes (Figure 1). Although the canonical and alternate pathways have been investigated individually, it is unknown whether these pathways function independently, or in combination.

It is now clear that masculinisation during human fetal development is dependent on both the canonical and alternate androgen biosynthesis pathways [54]. Confirming the importance of the alternate pathway, Flück et al. demonstrated that mutations to enzymes specific for the alternate pathway (AKR1C2/4) show perturbed sexual development resulting in DSD, even when the canonical pathway remains intact [52,54,55]. Further, O’Shaughnessy et al. recently showed that androsterone, an alternate pathway androgen, was a major precursor found in human male fetuses, with significantly higher circulating concentration compared to females [56]. This study also demonstrated that during human fetal development, the alternate pathway predominantly functions in peripheral nongonadal tissues including the placenta, liver and adrenal, indicating that tissues other than the testes contribute to masculinisation [56]. Masculinisation gender differences during fetal development due to the alternate pathway is likely due to *CYP17A1* expression, with expression occurring in the testis and adrenal [56]. Taken together, these studies indicate that the alternate pathway is important in male physiology, as evidenced by its requirement during key developmental timepoints. As the conversion of testosterone to the more potent DHT is important for the development of external genitalia [57], the non-developed male genitalia in human HSD17B3 deficient individuals suggests that the alternate pathway alone cannot compensate for the canonical pathway during fetal life. Therefore, both the canonical and alternate pathways are required for appropriate male sexual development, however, the specific interactions are unclear.

As DHT is largely responsible for the virilisation of external tissues, the alternate pathway may be responsible for the virilisation of tissues during puberty, rather than the canonical pathway. This is a possible explanation for the virilisation observed in HSD17B3 deficient humans during puberty, as it does not require testosterone to be produced, however this hypothesis has not been tested.

Whether the alternate pathway compensates for the canonical pathway in the absence of HSD17B3 in mice is currently unknown. The increased DHT production in HSD17B3 KO male mice may suggest the alternate pathway could be compensating for the canonical pathway to promote androgen action, by maintaining peripheral levels of DHT that can support normal androgen function [34]. Unfortunately, alternate pathway androgen precursors were not measured in either model so this theory cannot be confirmed at this stage. Then again, the increased levels of DHT [34] could be a result of the increased levels of circulating testosterone [17,34] from the canonical pathway, being converted into DHT. Regardless of whether the alternate pathway is compensating or not, this does not explain the continued testosterone production in the HSD17B3 KO mice and their ability to reproduce [17,34].

### 4.4. Other 17β-hydroxysteroid Dehydrogenases: Implications for Testosterone Synthesis

The HSD17B family consists of multiple subtypes that either reduce and/or oxidise precursors to convert them into other steroids. The amino acid sequence of reductive HSD17B enzymes is generally well conserved across human and mouse species, indicating that they originated from a common ancestor (Table 1). Therefore, it seems likely that some HSD17B enzymes may have overlapping functions and could be candidates to be compensatory mechanisms during the loss of HSD17B3. The homology of protein sequences has been assessed using protein BLAST (https://blast.ncbi.nlm.nih.gov (accessed on 26 October 2022)) to compare the amino acid sequence of mouse HSD17B3 with other reductive mouse HSD17B enzymes (Table 2).

Current literature dictates that in the testis, the canonical pathway of androgen production employs HSD17B3 as the critical enzyme for androstenedione reduction to testosterone. However, there are other known HSD17B enzymes capable of performing this reaction, including; HSD17B1 [64], HSD17B5 [28] and HSD17B12 [63]. Although these enzymes can carry out this function, it is with less efficiency compared to HSD17B3 [65].

HSD17B1 is viewed largely as an estrogenic enzyme, with its major function being the enzyme that catalyses the conversion of estrone to the more potent estrogen, estradiol. However, it has also been shown to have testosterone synthesising capabilities [66]. While HSD17B3 in Leydig cells is considered as the predominant enzyme responsible for testosterone synthesis in males, the situation is more complex during fetal testis development with both fetal Sertoli cells and fetal Leydig cells being required for testosterone synthesis [64]. In the fetal mouse testis, there is increasing evidence supporting that HSD17B1 and HSD17B3 both contribute to testicular testosterone production [18,47,64,66]. Human HSD17B1 can also convert androstenedione into testosterone, however, is less efficient compared to the mouse enzyme [47,67]. Virilisation during fetal development in the HSD17B3 KO male mice could be a result of the fetal Sertoli cells expressing HSD17B1, and as mouse HSD17B1 is more efficient at synthesising testosterone compared to human HSD17B1, this may partly explain the differences in the phenotype amongst the species [64,66]. Whether androgen action as a result of HSD17B1 activity is also true for the human fetal testis is unclear, however there is evidence for the preferential expression of *HSD17B1* and *HSD17B3* in human fetal Sertoli cells compared to Leydig cells [47]. While HSD17B1 is expressed in the mouse testis during fetal development, this expression is reduced to undetectable levels in adulthood [17,18,59,66]. Hakkarainen et al. demonstrated that HSD17B1 is required for normal steroid synthesis and spermatogenesis [66]. This study showed that in HSD17B1 KO male mice, HSD17B3 is up-regulated to compensate at 1 day postpartum and interestingly, continues to be up-regulated at 3 months of age (when HSD17B1 is normally undetectable) [66], likely to maintain testosterone production.

HSD17B5 is a multi-functional aldo-keto reductase enzyme that is involved in androgen, estrogen, progestin, and prostaglandin synthesis [68]. HSD17B5 is undetectable in the mouse testis and shows low expression in the human testis, yet is highly expressed in peripheral tissues [18,69]. While *Hsd17b5* was undetectable in the testis of HSD17B3 KO mice [17], its abundant expression in the periphery suggests HSD17B5 could be contributing to testosterone synthesis across multiple tissues as previously discussed. However, as a large proportion of testosterone in HSD17B3 KO adult mice is likely derived from the testis [34], HSD17B5 may be accompanied by other enzymes.

HSD17B12 is ubiquitously expressed throughout the body, including in the Leydig cells, throughout all life stages and is considered as a multi-functional enzyme predominantly involved in estradiol synthesis and the elongation of very long chain fatty acids [70]. Alongside HSD17B12’s major functions, mouse HSD17B12 has been shown in vitro to convert androstenedione to testosterone, and HSD17B12 in the Japanese eel synthesises 11-ketotestosterone, a predominant androgen in fish, from 11-ketoandrostenedione, suggesting that this function is conserved across some species [71]. However, human HSD17B12 is less efficient at converting androstenedione into testosterone [58,63]. The human HSD17B12 protein is highly specific for reducing estrone to estradiol, however, a single amino acid change of a bulky phenylalanine to a smaller leucine in mouse HSD17B12 impacts the binding site, making mouse HSD17B12 less specific, allowing more substrates (including androstenedione) to enter the active site [63]. This amino acid difference could explain the phenotypic differences seen between mouse and human. Consistent with the suggestion from Rebourcet et al. that mouse HSD17B12 could be compensating for the loss of HSD17B3 activity in HSD17B3 KO males [17], mice heterozygous for HSD17B12 have reduced levels of circulating testosterone, however, this could be a biproduct as a result of HSD17B12’s other functions [70]. *Hsd17b12* transcript expression in the testes of HSD17B3 KO males are slightly elevated [17]. However, as androstenedione and testosterone can be converted into estrogens by the enzyme aromatase (*Cyp19a1*), and because circulating androstenedione and testosterone is increased in HSD17B3 KO mice, this slight increase in *Hsd17b12* could also be due to a heightened conversion of estrogens, which is HSD17B12’s primary function.

Other reductase enzymes including mouse HSD17B6 and HSD17B7 have not been shown to convert androstenedione into testosterone, however, their ability to produce testosterone cannot be ruled out and would require further investigation. There is also the possibility that there are undiscovered HSD17B enzymes in the mouse that could be catalysing the reaction forming testosterone.

## 5. Concluding Remarks and Future Perspective

HSD17B3 has long been thought to be the main testosterone biosynthetic enzyme in adult males and required for sexual development and fertility. *HSD17B3* loss-of-function mutations in humans results in perturbed male sexual differentiation [22], however, recent ground-breaking studies have revealed that ablation of the canonical pathway in transgenic mice deficient in HSD17B3, has little impact on male development and fertility [17,34]. These HSD17B3 KO mouse models demonstrate the presence of species differences in androgen production and highlight the challenges that scientists experience when trying to study human disorders. While mice function similarly to humans, future studies involving the modelling of human androgen-related disorders in mice must acknowledge and consider the species differences.

The preservation of testosterone production and fertility in mice lacking HSD17B3 highlights the complexity of androgen biosynthesis and that the current view of mouse testosterone production is incomplete. The phenotypic differences observed between mice and humans with HSD17B3 deficiency could be a result of one or a combination of mechanisms, such as adrenal or other peripheral tissue androgen production, alternate pathway compensation, or testosterone production due to other HSD17B enzymes. The identification of compensatory mechanisms for androgen production is required and could provide novel insights into how androgens, specifically testosterone, are synthesised, and this will allow for more accurate models of human androgen-related disorders to be developed.

## Figures and Tables

**Figure 1 ijms-23-15555-f001:**
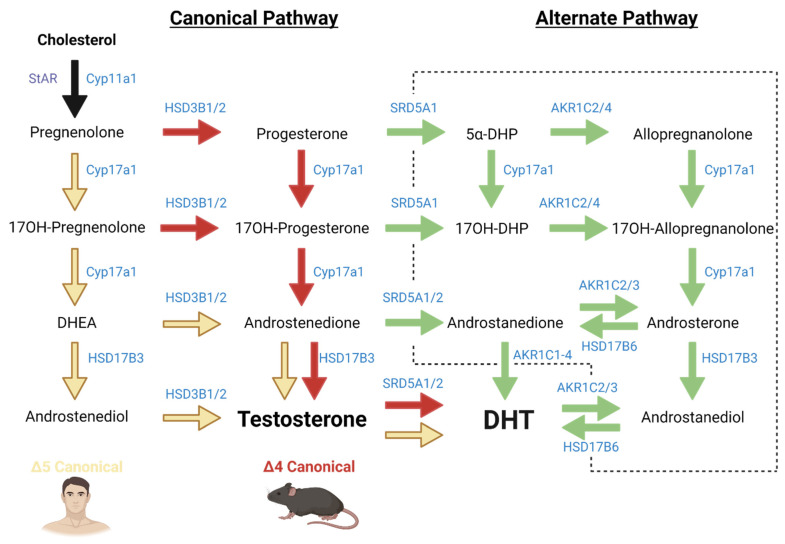
The current understanding of the canonical and alternate pathways of androgen biosynthesis. Independent of which pathway is used (black arrow), all androgens originate from cholesterol and are converted by multiple enzymes to produce the active androgens testosterone and dihydrotestosterone (DHT). The canonical pathway produces testosterone, which can act directly on the androgen receptor or be used as a precursor to the more potent androgen DHT. The alternate pathway (green arrows enclosed by the dotted line) can synthesise DHT, bypassing the need for testosterone synthesis. The canonical pathway includes Δ4 and Δ5 pathways; humans favour the Δ5 pathway (yellow arrows), whereas mice preference the Δ4 pathway (red arrows). Created with BioRender.com (accessed on 31 October 2022).

**Table 1 ijms-23-15555-t001:** Amino acid sequence homology of HSD17B enzymes between human and mouse.

HSD17B Subtype	Amino Acid Sequence Homology (%)	Preferred Functions
HSD17B1	68.29	Estrogen synthesis E1 → E2
HSD17B3	73.2	Androgen synthesis A4 → T
HSD17B5 (Human AKR1C3) (Mouse AKR1C6)	75.54	Prostaglandin synthesis
HSD17B7	78.01	Cholesterol synthesis Estrogen synthesis E1 → E2
HSD17B9 (Mouse HSD17B6)	47.17	Retinoid metabolism
HSD17B12	81.09	Fatty acid elongation Estrogen synthesis E1 → E2

E1 = Estrone, E2 = Estradiol, A4 = Androstenedione, T = Testosterone.

**Table 2 ijms-23-15555-t002:** Amino acid sequence homology of mouse HSD17B3 with other mouse HSD17B enzymes.

Mouse HSD17B Subtype	Amino Acid Sequence Homology Compared to Mouse HSD17B3 (% Identical)	Reductive Capacity	Shown to Convert A4 → T
HSD17B1	24.26	Yes [58]	Yes [59,60]
HSD17B5 (Mouse AKR1C6)	No significant similarity found	Yes [44]	Yes [28,44]
HSD17B7	26.85	Yes [61]	N/A
HSD17B9 (Mouse HSD17B6)	28.79	Yes [62]	N/A
HSD17B12	40.27	Yes [58]	Yes [63]

A4 = Androstenedione, T = Testosterone.

## Data Availability

Schematic of androgen production pathway Created with BioRender.com, agreement number KK24L9P31C. Amino acid sequence comparison was determined using NCBI protein blast suite (https://blast.ncbi.nlm.nih.gov (accessed on 26 October 2022)).

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
