# Peer review of "New Insights into Testosterone Biosynthesis: Novel Observations from HSD17B3 Deficient Mice"

_ijms, 2022, doi:10.3390/ijms232415555_

Round 1

Reviewer 1 Report

I found that this paper to be overall very well informed, objective and well written, however certain points need to be clarified before the manuscript could be considered for publication.

·        Tables are not satisfactory or enough. More tables are needed to compare between the previous studies.

·        Future perspective should be provided.

Author Response

The authors thank reviewer 1 for their comments on the manuscript and find them beneficial for the manuscript. In line with all suggestions, amendments have been made.

Table 1 has been updated to include the preferential function of each enzyme and table 2 has been updated to indicate whether the enzymes have previously been shown to produce testosterone. We believe that this adds to the tables and the manuscript making them satisfactory.

Future perspective has been included on lines 408-412 and the sentence on from lines 419-522 has been revised to also indicate what is needed in the future.

Reviewer 2 Report

This is a very well-written and informative review on the regulation of testosterone biosynthesis. The authors may wish to consider the following minor points:

1. Line 4: the end of the author line seems incomplete.

2. Lines 32-35: Since the authors are referring to the human context alone, perhaps it would be better to rephrase the last sentence to read "at any point during life can result in complications contributing to lower
quality of life and premature death in men."

3. Lines 81-83: This sentence lacks supporting references.

Author Response

The authors thank reviewer 2 for their comments on the manuscript. In line with all suggestions, amendments have been made.

Author list has been updated.

Reviewer 2 suggested to rephase sentence on lines 32-35 and their suggestion was accepted, and the manuscript has been changed to show this.

Supporting references have been added to sentence on lines 81-83.

Round 2

Reviewer 1 Report

The manuscript is greatly improved. The manuscript could be considered for publication.